# Association between Non-Restorative Sleep and Quality of Life in Chinese Adolescents

**DOI:** 10.3390/ijerph17197249

**Published:** 2020-10-04

**Authors:** Ningjing Chen, Daniel Yee Tak Fong, Sha Li, Janet Yuen Ha Wong

**Affiliations:** School of Nursing, Li Ka Shing Faculty of Medicine, The University of Hong Kong, 21 Sassoon Road, Hong Kong 999077, China; chenningjing@connect.hku.hk (N.C.); joylisha@connect.hku.hk (S.L.); janetyh@hku.hk (J.Y.H.W.)

**Keywords:** non-restorative sleep, quality of life, adolescents, Chinese

## Abstract

To examine non-restorative sleep and its impact on quality of life (QOL) in Chinese adolescents, this cross-sectional study included 2827 students aged 12–20 who were selected from 15 secondary schools in Hong Kong, China. Non-restorative sleep was assessed by a single item, rated on a 0–10 scale: “To what extent did you feel refreshed upon awakening over the past month?”. QOL was evaluated by the World Health Organization Quality of Life Measure—Abbreviated Version Hong Kong Chinese Version. Univariable and multivariable linear mixed-effects regressions were conducted to examine the influence of non-restorative sleep (NRS) on QOL. In the present sample, the mean overall QOL was 13.83, and the extent of feeling refreshed upon awakening was 4.75 on average. In multivariable linear mixed-effects regression, one unit feeling less refreshed upon awakening was associated with 0.37 units of poorer overall QOL after adjusting for age, gender, medical conditions, parental education and occupation, weekly hours of aerobic exercises, smoking and drinking habits. Additionally, adolescents with a significantly poorer overall QOL were more likely to be older, have medical problems, have parents with an educational level of primary school or below, have an unemployed father, engage in less weekly aerobic exercise, and be current smokers or former drinkers. Adolescents who experienced non-restorative sleep had a poorer QOL. Future studies are needed to alleviate non-restorative sleep to improve health outcomes in adolescents.

## 1. Introduction

Non-restorative sleep (NRS) refers to experiencing an unrefreshed feeling upon awakening [1]. It is not the same as sleep quality, but it is identified as an insomnia symptom in the International Classification of Sleep Disorders [2]. However, it is not always accompanied by other insomnia symptoms such as difficulties falling and staying asleep, which means that even individuals who have a normal sleep duration may experience NRS [3]. The specific predictors for NRS remain unclear [1]. However, recent studies have shown that a younger age, smoking, less exercise, anxiety and depression are associated with the occurrence of NRS [3,4]. Adolescence is the life period that ranges from 10 to 24 years of age [5]. Due to variations in population characteristics, sample sizes and measurements, the prevalence of NRS in adolescents has been shown to vary from 4.0 to 11.7% [3,6,7].

NRS is a consequence of sleep disturbance and reflects the process of human bodies recovering from physical and mental workload being attenuated [1]. Individuals who feel less refreshed upon awakening are more vulnerable to fatigue, daytime sleepiness and cognition impairment in daily life [1]. Additionally, NRS heightens the risk for mental disorders [8], as it can deteriorate psychological functioning and social relationships. People who experience NRS are also at a higher risk of environmental accidents such as not remembering to turn off taps or gas, unintentional falls and injuries [9]. Thus, NRS can negatively impact quality of life (QOL), which refers to “an individual’s perception of their position in life in the context of the culture and value systems in which they live and in relation to their goals, expectations, standards and concerns [10]”.

QOL comprises individuals’ physical, psychological, social and environmental health [11], and adolescents’ QOL has been one of the most important research priorities worldwide [12]. Adolescence is a life stage characterized by rapid physical and psychological development, during which individuals facing various challenges [13]. Since adolescents’ capacities have not been developed sufficiently, they are vulnerable to environmental health threats and the health consequences may last in their later life periods [13]. More than 3000 adolescents die per day due to traffic injuries, suicide, or violence, and approximately 10–20% of adolescents suffer from psychological dysfunction globally [14]. Adolescents with mental health problems are at increased risk of discrimination and social withdrawal [15]. Examining the relationships between QOL and its predictors may yield critical strategies to improve individuals’ QOL. Previous research revealed that regular exercise [16], no medical problems [17] and parents with higher educational levels [18,19] were positively associated with adolescents’ QOL, whereas older age [20], female gender [18], unhealthy lifestyles such as smoking and alcohol consumption [21], having an unemployed father [21] and sleep disturbance [22] were negatively associated with adolescents’ QOL.

Although these studies mentioned above provide evidence that NRS affects QOL [1,8,9], the majority of them only measured one or two facets of QOL or did not adjust for other covariates, such as exercise, smoking and drinking habits, which were associated with QOL [16,21]. Hence, the relationship between NRS and QOL may fluctuate. Further, previous studies on NRS have mainly focused on adults [23,24], whereas little attention has been paid to adolescents [3,6,7]. Moreover, the association between NRS and QOL in adolescents remains unclear. Although some studies have shown that sleep debt and disturbed sleep are deleterious to adolescents’ physical and psychological health [22,25], NRS is different from sleep debt and disturbed sleep based on its definition [1,3]. Therefore, as NRS in adolescents and its impact on QOL need further investigation, this study aimed to examine NRS and its impact on QOL in Chinese adolescents.

## 2. Material and Method

### 2.1. Study Design and Participants

This cross-sectional school study was conducted in Hong Kong between April and July 2016. We took data from a study that assessed health problems and noise exposures in school students. Students in grades 7–12 were recruited, and those who could not read Chinese were excluded. The sample size calculation was conducted to reliably estimate the prevalence of health problems and noise exposures. We considered two scenarios, representing a moderate and low prevalence. In scenario 1, we conservatively took 50% prevalence for a maximum 5% error, which resulted in 385 students with a 95% confidence interval (CI). In scenario 2, we took 5% prevalence for a maximum error of 1%, which resulted in 1825 students. Hence, the targeted number of students was 2000. This study assessed the impact of NRS on QOL and 10 factors for adjustment. Using the conventional rule of thumb of 10 subjects per factor, 100 subjects were needed. Therefore, a sample of 2000 students was adequate.

This study received ethical approval from the Institutional Review Board of the University of Hong Kong/Hospital Authority Hong Kong West Cluster. This study was conducted in accordance with the Declaration of Helsinki. We randomly selected one school from each of the 18 districts in Hong Kong and sent letters inviting these schools to participate in our study; 15 schools agreed to be included. Before participants enrolled in our study, participants and their parents were informed of the study details and were asked to sign a consent form.

### 2.2. Measurements

The following battery of questionnaires was self-administered.

#### 2.2.1. Non-Restorative Sleep

NRS was measured by the item “To what extent did you feel refreshed upon awakening over the past month?”. Participants responded on a scale ranging from 0 to 10, where 0 = “did not feel refreshed at all” and 10 = “felt fully refreshed” [24]. It was then reverse coded so that a higher NRS score indicated more severe NRS.

#### 2.2.2. Quality of Life

The World Health Organization Quality of Life Measure–Abbreviated Version (WHOQOL-BREF) is widely used to evaluate young cohorts’ QOL in cross-cultural settings [11]. In this study, QOL was assessed by the WHOQOL-BREF Hong Kong Chinese Version, which has been shown to have good psychometric properties [26]. This version of the WHOQOL-BREF is divided into four domains—physical health, psychological, social relationships, and environment—and another two items assess individuals’ general QOL and health perception [11]. Each domain score was generated as the mean score of all items within the domain and was transformed into a 4–20 scale. The mean score of these four domains was used to calculate overall QOL score and higher scores reflected a better QOL.

#### 2.2.3. Sociodemographic Characteristics

We collected the participants’ sociodemographic information, including age, gender, medical conditions (blood diseases, attention deficit hyperactivity disorder, heart disease, obesity, anxiety disorders, depressive disorders, eating disorders, scoliosis, respiratory diseases and other diseases), parental educational levels (primary school or below, secondary school/diploma/associate degree, tertiary or above, not applicable), and parental occupation (employed, self-employed, employer, housekeeping, job-seeking, retired, student, not applicable).

#### 2.2.4. Lifestyle

Participants’ lifestyle data included hours of aerobic exercise per week, smoking habit (never, former smoker, current smoker) and alcohol consumption habits (never, former drinker, current drinker).

### 2.3. Statistical Analysis

Given the possible extra-covariance among participants recruited from the same school, linear mixed-effects regression was used to adjust for clustering by school. Previous research suggested individuals’ sociodemographic and lifestyle characteristics including age [20], gender [18], medical problems [17], parents’ educational levels [18,19], parents’ occupations [21], exercise habits [16], smoking and alcohol consumption [21] were associated with adolescents’ QOL. Therefore, these variables were included as covariates in this study. A univariable linear mixed-effects regression was conducted to examine the influence of NRS, sociodemographic and lifestyle characteristics on QOL when no adjustment was considered. A multivariable linear mixed-effects regression was conducted to estimate the adjusted effects. Model adequacy was obtained by examining the standardized residuals, and the presence of multicollinearity was assessed by tolerance. Each estimate was accompanied by a 95% CI where appropriate. In all significance tests, a nominal level of 5% was considered statistically significant. The descriptive analyses were performed using SPSS version 25.0 (SPSS Inc, Armonk, NY, USA) and the linear mixed-effects regressions were conducted in RStudio-1.2.5042 using lmerTest package [27].

## 3. Results

### 3.1. Sociodemographic and Lifestyle Characteristics

A total of 2827 students participated in our study. Table 1 shows their sociodemographic and lifestyle information. Participants’ mean age was 15.24 years (range: 12–20 years, SD = 1.59 years). Nearly 900 participants (32.2%) had at least one doctor-diagnosed medical condition. The majority of participants’ fathers (61.1%) were currently employed, while only 3.7% were retired, and their mothers were mostly either employed (47.3%) or housekeepers (35.5%). On average, participants engaged in 2.90 h (SD = 3.15 h) of aerobic exercise per week. Less than 3% of all participants were former or current smokers, whereas more than 10% were current alcohol drinkers.

### 3.2. NRS and QOL

Table 2 shows participants’ scores for NRS, each QOL domain, and overall QOL. The mean NRS score was 4.75 (SD = 2.30). In the total sample, 4.5% (128) reported they did not feel refreshed at all upon awakening, whereas 2.9% (81) felt fully refreshed. Regarding QOL, participants’ highest scores were generally in the physical health domain (mean = 14.33, SD = 2.37), whereas the lowest scores were in the psychological domain (mean = 13.47, SD = 2.81). The mean overall QOL score was 13.83 (SD = 2.30).

### 3.3. Impact of NRS on QOL

Table 3 shows the impact of NRS on each QOL domain after adjusting for sociodemographic and lifestyle characteristics. Feeling less refreshed upon awakening was significantly associated with a poorer QOL in each domain (all *p* < 0.001). One unit of feeling less refreshed upon awakening was associated with a decrease of 0.37 units in overall QOL (95% CI = −0.41 to −0.34), 0.42 units in the physical health domain (95% CI = −0.46 to −0.39), 0.46 units in the psychological domain (95% CI = −0.50 to −0.41), 0.25 units in the social relationships domain (95% CI = −0.30 to −0.20), and 0.35 units in the environment domain (95% CI = −0.39 to −0.31).

### 3.4. Factors Associated with Overall QOL

Table 4 shows factors associated with overall QOL. In all models, residuals did not deviate extremely from the assumption of normality. Given that the tolerance ranges for the educational levels of participants’ fathers and mothers were 0.39–0.44 and 0.39–0.46, respectively, they were individually entered in the multivariable model. The tolerance of the resulting multivariable models ranged from 0.84 to 0.99, which did not indicate multicollinearity.

In multivariable analyses, students with significantly lower overall QOL were more likely to be older, have medical problems, have parents with an educational level of primary school or below, have an unemployed father, participate in less weekly aerobic exercise, and be a current smoker or former drinker.

## 4. Discussion

To our knowledge, this was the first study to use a large representative sample to examine NRS and its impact on QOL in Chinese adolescents. After adjusting for age, gender, medical history, parental education and occupation, hours of aerobic exercise per week, smoking and drinking habits, feeling less refreshed upon awakening was associated with a reduced QOL in all aspects including physical health, psychological, social relationships, environment and the overall QOL.

On average, Chinese adolescents reported 47.5% of feeling unrefreshed upon awakening, which was slightly higher than the corresponding 35.0% reported in a previous household study among adults in Hong Kong [24]. This discrepancy could be explained in part by age differences [4], as participants recruited in our study were much younger on average (15.24 vs. 32.00 years). Age is inextricably linked to circadian rhythms, which somewhat determine the timing of falling asleep and awakening, as well as the alertness level while awake. As a result of circadian systems, adolescents experience a delayed sleep phase pattern, hindering them from falling asleep until late. However, the time for class is often earlier than the time when young students would naturally awaken, which means that students have to wake up at undesirable sleep points [25]. Therefore, young students would generally feel less awake than the adults in the previous household study.

Participants in this study had lower mean scores in each QOL domain, relative to general adolescents in a Hong Kong territory-wide survey conducted in 2014−2015 (14.33 vs. 16.30, 13.47 vs. 15.20, 13.82 vs. 15.20 and 13.71 vs. 15.40 in the physical health, psychological, social relationships and environment domain, respectively) [28]. The prevalence rates of “very poor sleep quality” and “very good sleep quality” in the general Hong Kong adolescent population were found to be 0.4 and 13.2%, respectively. However, in our study, 4.5% of the participants did not feel refreshed after sleep, whereas only 2.9% of the participants felt fully refreshed upon awakening. Therefore, our participants tended to have poorer sleep quality than the general Hong Kong adolescent population. Poor sleep quality negatively impacts adolescents’ QOL [22,25], which might explain the differences in QOL between these two studies. Prior research also indicated that adolescents’ QOL could be improved if their family income increased [18]. In contrast, although participants in our study were thought to have a better economic status, their QOL in each domain was still poorer than that of Brazilian teenagers living in undeveloped areas [18] and young residents from rural areas in mainland China [29]. This might be due to culture differences and their effects on self-perceptions, since QOL depends on individuals’ perceptions relevant to their culture and living environment. Among all QOL domains, our study also showed that Hong Kong secondary school students scored lowest in the psychological domain, which was consistent with a large-scale school-based study in French adolescents [21]. These findings indicate that young school students’ psychological health may need further attention and targeted interventions to improve their psychological functioning are desired.

Our results indicate that feeling less refreshed after sleep is negatively associated with all QOL domains, which is in line with prior studies on the effects of sleep quality on QOL in adolescents [22,25,30,31]. NRS prevents the human body from functioning optimally, causes daytime sleepiness, cognitive impairment, and mood disturbances, and jeopardizes daily performance in studying, working, and engaging in entertainment [1]. Thus, students who experienced NRS reported a poorer QOL. Our results therefore provide some implications for medical practice. For example, from a clinical perspective, treating NRS may be a novel consideration for improving QOL.

In our study, adolescents with an unemployed father tended to have a poorer QOL. This is consistent with findings from large-scale school-based studies in French and Slovakian adolescents [21,32]. Often, the father provides the primary income in a family. If an adolescent’s father is unemployed, their household income will most likely be reduced, resulting in a poorer QOL [18]. However, unemployed mothers may have more time to interact with their children, such as sending them to and from school and assisting with their homework [33]. This might be the reason why mothers’ unemployment did not have a statistically significant effect on their children’s QOL.

The observed effects of parental educational levels in this study were in line with other research findings regarding adolescents’ well-being [18]. Parents with higher educational levels tend to have better knowledge attainment and cognitive functioning. They may pay more attention to their children’s development and provide more care when necessary, including giving feedback on homework and encouraging children to enroll in extracurricular learning activities. Hence, adolescents whose parents had a tertiary educational level or above had a perceived better overall QOL than their peers whose parents had a primary school education or below.

The effects of age, medical conditions, and exercise habits in Chinese youths’ health and well-being could be applicable to other contexts [16,18,20]. Interestingly, in a multivariable analysis, being a former smoker or current drinker were non-significant predictors of a poorer QOL, which was inconsistent with prior research indicating that smoking or drinking had inverse effects on adolescents’ QOL [21,34]. However, although former smokers were more likely to experience physical and psychological disorders, their likelihood of engaging in risky health behaviors was comparable with that of non-smokers [35]. Additionally, former smokers spent more time exercising per week than non-smokers (3.74 vs. 2.89 h). Thus, differences in QOL between former smokers and non-smokers were possibly statistically insignificant. Similarly, although alcohol consumptions has been suggested to impair attentional function, verbal learning, and memory [36,37]. Adolescents often regard drinking as a way to strengthen relationships with peers and establish new friendships [38]. Under such circumstances, adolescents may not realize the adverse health consequences of alcohol consumption. QOL is based on self-conceptions, which might explain why the differences in QOL between current drinkers and non-drinkers did not reach statistical significance.

There were several limitations to this study. First, NRS was measured by a single item rather than a validated multi-item scale that might be more sensitive. The 12-item Traditional Chinese Non-restorative Sleep Scale [39] and its short nine-item version [40] have shown satisfactory reliability and validity in adults; however, there has been no confirmed NRS assessment tool for adolescents. Nevertheless, a single-item scale is appealing in minimizing administrative burdens in such a large sample. Second, we assessed NRS by self-report, which might be subject to recall bias. Potential objective assessments of NRS are available, such as electroencephalography; however, the specificity and sensitivity have not been well tested [41]. Therefore, it is unlikely to be feasible in a large-scale survey. Third, the cross-sectional design of this study does not enable the establishment of causality due to possible reverse causation. A longitudinal study with follow-up for QOL measurements will be desirable to assess the temporal association. Fourth, we did not include other covariates that may confound the association between NRS and QOL, such as household crowding, noise levels, sleep dissatisfaction, and the frequency of having a bad night’s sleep. Fifth, we have not delineated the potential difference of NRS between school days and non-school days. Future studies may examine if school days would be associated with NRS and also QOL.

## 5. Conclusions

Adolescents appear to suffer NRS more than adults, with a clear association between NRS and QOL in adolescents. As NRS may not co-exist with other sleep problems, it is a sleep issue that deserves attention for improving the QOL of young generation. Future longitudinal studies could be conducted to help establish the causal relationship between NRS and QOL.

## Figures and Tables

**Table 1 ijerph-17-07249-t001:** Sociodemographic and lifestyle characteristics of the 2827 Chinese adolescents.

Items	Mean ± SD/*n* (%)
Sociodemographic characteristics	
Age (years, 10 missing, 0.4%)	15.24 ± 1.59
Gender (74 missing, 2.6%)	
Female	1565 (55.4)
Male	1188 (42.0)
Medical conditions (100 missing, 3.5%)	
Yes	909 (32.2)
No	1818 (64.3)
Father’s educational levels (92 missing, 3.3%)	
Primary school or below	312 (11.0)
Secondary school/diploma/associate degree	1590 (56.2)
Tertiary or above	559 (19.8)
Not applicable	274 (9.7)
Mother’s educational levels (74 missing, 2.6%)	
Primary school or below	343 (12.1)
Secondary school /diploma/associate degree	1710 (60.5)
Tertiary or above	475 (16.8)
Not applicable	225 (8.0)
Father’s occupation (93 missing, 3.3%)	
Employed	1728 (61.1)
Self-employed	241 (8.5)
Employer	220 (7.8)
Housekeeping	52 (1.8)
Job-seeking	57 (2.0)
Retired	105 (3.7)
Student	1 (0.1)
Not applicable	330 (11.7)
Mother’s occupation (76 missing, 2.7%)	
Employed	1337 (47.3)
Self-employed	96 (3.4)
Employer	84 (3.0)
Housekeeping	1004 (35.5)
Job-seeking	34 (1.2)
Retired	9 (0.3)
Student	1 (0.1)
Not applicable	186 (6.6)
Lifestyle characteristics	
Weekly aerobic exercises (hours, 122 missing, 4.3%)	2.90 ± 3.15
Smoking (15 missing, 0.5%)	
Never	2733 (96.7)
Former smoker	57 (2.0)
Current smoker	22(0.8)
Alcohol drinking (16 missing, 0.6%)	
Never	2431(86.0)
Former drinker	56(2.0)
Current drinker	324(11.5)

SD: standard deviation.

**Table 2 ijerph-17-07249-t002:** Non-restorative sleep and quality of life of the 2827 Chinese adolescents.

Items	Mean ± SD
Non-restorative sleep (range: 0–10, 69 missing, 2.4%)	4.75 ± 2.30
Domains (range: 4–20)	
Physical health (7 missing, 0.25%)	14.33 ± 2.37
Psychological (6 missing, 0.21%)	13.47 ± 2.81
Social relationships (5 missing, 0.18%)	13.82 ± 2.95
Environment (1 missing, 0.04%)	13.71 ± 2.60
Overall QOL (17 missing, 0.60%)	13.83 ± 2.30

SD: standard deviation; CI: confidence interval; QOL: quality of life.

**Table 3 ijerph-17-07249-t003:** The impact of non-restorative sleep on each domain of quality of life.

Domains	Univariable Analysis				Multivariable Analysis ^a^			
	*n*	Estimate	95% CI	*p*	*n*	Estimate	95% CI	*p*
Physical health	2751	−0.44	−0.48 to −0.41	<0.001	2445	−0.42	−0.46 to −0.39	<0.001
Psychological	2753	−0.48	−0.52 to −0.43	<0.001	2451	−0.46	−0.50 to −0.41	<0.001
Social relationships	2754	−0.25	−0.30 to −0.21	<0.001	2449	−0.25	−0.30 to −0.20	<0.001
Environment	2757	−0.37	−0.41 to −0.33	<0.001	2452	−0.35	−0.39 to −0.31	<0.001
Overall QOL	2743	−0.39	−0.42 to −0.36	<0.001	2441	−0.37	−0.41 to −0.34	<0.001

CI: confidence interval; QOL: quality of life. ^a^ The results were obtained after adjusting for age, gender, medical conditions, parental educational levels and occupation, exercise habits, smoking and drinking habits.

**Table 4 ijerph-17-07249-t004:** Factors associated with Chinese adolescents’ overall quality of life.

Variable	Univariable Analysis			Multivariable Analysis		
	Estimate	95% CI	*p*	Estimate	95% CI	*p*
Non-restorative sleep (range: 0–10, *n* ^a^ = 2743)	−0.39	−0.42 to −0.36	<0.001	−0.37	−0.41 to −0.34	<0.001
Age (years, *n* ^a^ = 2800)	−0.30	−0.35 to −0.24	<0.001	−0.20	−0.25 to −0.15	<0.001
Gender (ref. Male, *n* ^a^ = 2736)						
Female	−0.08	−0.27 to 0.10	0.377	−0.03	−0.20 to 0.13	0.688
Medical conditions (ref. Yes, *n* ^a^ = 2711)						
No	0.53	0.35 to 0.71	<0.001	0.45	0.29 to 0.62	<0.001
Father’s education ^b^ (ref. Primary school or below/not applicable, *n* ^a^ = 2720)						
Secondary school/diploma/associate degree	0.41	0.20 to 0.63	<0.001	0.23	0.02 to 0.44	0.036
Tertiary or above	0.60	0.32 to 0.89	<0.001	0.41	0.15 to 0.67	0.003
Mother’s education ^b^ (ref. Primary school or below/not applicable, *n* ^a^ = 2739)						
Secondary school/diploma/associate degree	0.33	0.11 to 0.55	0.004	0.12	−0.09 to 0.33	0.253
Tertiary or above	0.43	0.14 to 0.73	0.004	0.31	0.04 to 0.59	0.026
Father’s occupation (ref. Working ^c^, *n* ^a^ = 2721)						
Non-working ^d^	−0.48	−0.70 to −0.27	<0.001	−0.35	−0.55 to −0.14	<0.001
Mother’s occupation (ref. Working ^c^, *n* ^a^ = 2736)						
Non-working ^d^	−0.02	−0.20 to 0.15	0.798	−0.02	−0.18 to 0.14	0.845
Weekly aerobic exercises (hours, *n* ^a^ = 2691)	0.06	0.04 to 0.09	<0.001	0.05	0.02 to 0.08	<0.001
Smoking (ref. Never, *n* ^a^ = 2796)						
Former smoker	−0.84	−1.45 to −0.23	0.007	−0.50	−1.15 to 0.14	0.127
Current smoker	−1.66	−2.62 to −0.71	<0.001	−2.31	−3.70 to −0.92	0.001
Alcohol consumption (ref. Never, *n* ^a^ = 2795)						
Former drinker	−1.13	−1.74 to −0.53	<0.001	−0.77	−1.39 to −0.14	0.017
Current drinker	−0.41	−0.68 to −0.15	0.002	−0.03	-0.29 to 0.24	0.852

CI: confidence interval. ^a^ The sample size for the univariable analysis. In the multivariable analysis, the sample size was 2441 for the model without the educational levels of participants’ father (*n* = 2399) and mothers (*n* = 2409). ^b^ Given that the tolerance ranges for the educational levels of participants’ fathers and mothers were 0.39–0.44 and 0.39–0.46, respectively, they were individually entered in the multivariable model. ^c^ Working included employed, self-employed, and employer. ^d^ Non-working included housekeeping, job-seeking, retired, student and not applicable.

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
