# Peer review of "Association between Non-Restorative Sleep and Quality of Life in Chinese Adolescents"

_ijerph, 2020, doi:10.3390/ijerph17197249_

Round 1

Reviewer 1 Report

This is a well-written paper that investigates the well-researched question on the association between non-restorative sleep and quality of life in a distinct population: Chinese adolescents.  The authors give an excellent background of the topic and present their findings clearly, however they have made an error in their choice of statistical model and also not considered two very serious limitations.

Major points:

The authors need to use multilevel modeling to adjust for clustering by school.

The authors do not describe their method for covariate selection.  Did they base it on the univariate analysis presented in Table 4?  On a causal diagram?  On prior literature?  On a statistical (backward, forward) selection model?  On some combination of methods?  Did the authors consider the possibility that they might need to use different covariate sets for the different outcomes in Table 3?

The authors do not address the limitation that this is a cross-sectional study and there is a good chance of the observed association being due to reverse causation. 

The authors also need to address the limitation that they do not have data on and therefore cannot adjust for variables such as household crowding or noise that could impact QOL and therefore NRS (i.e., potential sources of reverse causation).

Minor points:

What does the 95% CI refer to in section 3.2 and Table 2?  Means should not have confidence intervals.

Did the authors check to make sure that the variables in Table 2 were normally distributed?  If not, they should present median (IQR) instead of mean (SD).

It would be clearer if, in section 3.3 and Table 3, the authors present their results as negative numbers, as the association is between increased NRS score and decreased QOL.

The authors should indicated the final n in their analytic sample.  Tables 1 and 2 indicates some missing data, and if Table 3 represents a complete case analysis, the n will be smaller than the original study sample.

Table 4 contains univariate associations, which are relevant to covariate selection (as mentioned above), as well as a multivariate prediction model. The reasons for running these analyses are not well-described in the text and the results need to be presented in the context of those reasons.

Reviewer 2 Report

The authors investigated the impact of non restorative sleep on the quality of life in chinese adolescents, accounting for various sociodemographic characteristics and lifestyle factors. First and foremost, the current study underlines the importance of including NRS in future sociodemographic and epidemiological studies. It offers an accurate observation focusing on parametrs of adolescents' quality of life, and compares the data acquired here to previous knowledge, offering possible explanations for the discrepancies observed. Most importantly, despite its stated limitations, it contributes specific evidence that link NRS with a lower quality of life in adolescents.

An addition that might strengthen the manusctipt as it stands, regards accounting also for the irregularity of the adolescent's sleep schedule, as well as bedtime (lights off/on). It would have been nice to see also other sleep-related parameters to be accounted for, such as global sleep dissatisfation, as well as how often a bad night's sleep occured, if such information are available to the authors.

Round 2

Reviewer 1 Report

The authors have satisfactorily responded to all major comments. However, they should remove the 95% CIs from section 3.2 and Table 2. I now see why I was having difficulty with the results presented in section 3.3 and Tables 3/4: the NRS variable is coded backward, with the highest score reflecting RESTORATIVE sleep, which is very confusing and not typical when the variable's name is NON-RESTORATIVE sleep. While it is accurate to leave the results as is, because the name of the variable does not match the high score, it is confusing to the reader. I suggest that the authors reverse code the variable (i.e., 10="did not feel refreshed at all") so that the highest score reflects NON-RESTORATIVE sleep; that way, the name of the variable and the interpretation of the results in Section 3.3 and Table 3 (and the top of Table 4) will make sense. When this is done, the betas will be negative.

Author Response

Please see the attachment. Many thanks!

Reviewer 3 Report

Thank you for coorecting the manuscript according to my suggestions.

Author Response

We thank you for your kind words.